# Miniaturized electromechanical devices with multi-vibration modes achieved by orderly stacked structure with piezoelectric strain units

Piezoelectric devices based on a variety of vibration modes are widely utilized in high-tech fields to make a conversion between mechanical and electrical energies. The excitation of single or coupled vibration modes of piezoelectric devices is mainly related to the structure and property of piezoelectric materials. However, for the generally used piezoelectric materials, e.g., lead zirconate titanate ceramics, most of piezoelectric coefficients in the piezoelectric matrix are equal to zero, resulting in many piezoelectric vibration modes cannot be excited, which hinders the design of piezoelectric devices. In this work, an orderly stacked structure with piezoelectric strain units is proposed to achieve all nonzero piezoelectric coefficients, and consequently generate artificially coupled multi-vibration modes with ultrahigh strains. As an example, an orderly stacked structure with two piezoelectric strain units stator, corresponding to 31–36 coupled vibration mode, was designed and fabricated. Based on this orderly stacked structure with two piezoelectric strain units stator, we made a miniature ultrasonic motor (5 mm$^{Length}$ × 1.3 mm$^{Height}$ × 1.06 mm$^{Width}$). Due to the ultrahigh strain of the 31–36 coupled vibration mode, the velocity per volume of the motor reached 4.66 s$^{-1}$ mm$^{-2}$. Furthermore, its moving resolution is around 3 nm, which is two orders higher than that of other piezoelectric motors. This work sheds a light on optimizing the performance of state-of-the-art electromechanical devices and may inspire new devices based on multi-vibration modes.

With the rapid development of micro-machinery, micro-manufacturing, minimally invasive robot, micro-electronics and other high-tech areas, the electromechanical devices with high motion speed, miniaturized size and high resolution are highly desired[1–4]. Compared with traditional electromagnetic motors/actuators, electromechanical devices possess the advantages of fast response, low noise, compact size and no electromagnetic interference, thus have attracted considerable attentions in last two decades[5–7]. They are widely used in aerospace[8,9], precision positioning systems[10–12], automatic zoom lens[1,13,14], bio-technology and medicine[15–17], and nano-/micro-electro-mechanical systems (NEMS/MEMS)[18–20]. In addition, due to the advantage that the output power density is almost independent of dimensional changes[21], piezoelectric devices are promising to simultaneously achieve miniaturization, nanoscale resolution and fast response[5,22].

Under the excitation of external electric fields, the piezoelectric materials can generate stresses and strains via inverse piezoelectric effect to make the piezoelectric materials vibrate at standing wave

e-mail: gaoxiangyu@xjtu.edu.cn; ful5@xjtu.edu.cn

frequency[23]. Utilization of the coupling of two orthogonal vibration modes is the most widely driving principle of piezoelectric motors, such as the coupling of the first longitudinal ($L_1$) mode and the second bending ($B_2$) mode[24–26]. However, to obtain high output velocity, this kind of devices tend to take the longest side as the driving end, making them difficult for miniaturization and integration. In addition, since there is a deadzone in ultrasonic devices as a result of its driven by friction, the displacement resolution is generally in submicron level[13,16,27,28]. For example, Li et al.[29] proposed a piezoelectric stator based on $L_1B_2$ mode with the length of 13.5 mm, and the maximum speed is about 101 mm s$^{-1}$, but the accuracy is only 0.3 μm. Therefore, the development of ultrasonic piezoelectric devices based on alternative vibration coupling modes is crucial for solving the issue to simultaneously achieve miniaturization and high output displacement. However, for piezoelectric ceramics, there are only five nonzero piezoelectric coefficients because of ∞m point group symmetry, resulting in that many vibration modes cannot be excited, which greatly hinders the design and application of piezoelectric devices[30].

Recently, it has been reported that all nonzero and ultrahigh piezoelectric coefficients can be realized in piezoelectric materials based on the construction idea of metamaterials[30,31], inspiring the design of electromechanical devices. However, many piezoelectric devices, such as ultrasonic motors, require coupled vibrational modes for high electromechanical conversion capability. In previous reports, the coupled vibration modes and the construction of all nonzero piezoelectric coefficients aren't achieved simultaneously, since the different piezoelectric strain units and/or different boundary conditions are adopted for inducing different piezoelectric vibration modes. For example, as reported in ref. 31, the non-zero piezoelectric coefficients $d_{14}$ and $d_{11}$ were achieved by triangle and square strain units, respectively. Thus, to fulfill the requirements of piezoelectric devices, the metamaterials that can achieve all non-zero piezoelectric coefficients and coupled vibration modes by the uniform piezoelectric strain unit and boundary condition are highly desired.

In this work, an orderly stacked structure based on piezoelectric strain units (OSSPSU) was proposed. According to the synergistic strain effect among the piezoelectric strain units, we achieved full artificial vibration modes corresponding to all nonzero, programmable and ultrahigh strains. Combined with the Finite Element Method

(FEM), we designed and fabricated an OSSPSU motor with the dimensions of 5 mm$^{Length}$ × 1.3 mm$^{Height}$ × 1.06 mm$^{Width}$ based on two [001]-poled Pb(In$_{1/2}$Nb$_{1/2}$)O$_3$-Pb(Mg$_{1/3}$Nb$_{2/3}$)O$_3$-PbTiO$_3$ (PIMNT) single crystal units, which couples 31- and 36-modes. To the best of our knowledge, it possesses the highest step accuracy (3 nm) among standing wave ultrasonic motors in open-loop control[1,21,32,33]. Due to the ultrahigh strain of the 31–36 coupled vibration mode of the OSSPSU stator, the newly proposed OSSPSU motor exhibits a large output speed in unit volume of 4.66 s$^{-1}$ mm$^{-2}$. The proposed OSSPSU is potential to be integrated into micromachined industrial systems currently in-use.

## Results
### Design of the OSSPSU
Here we show a method for achieving full artificial vibration modes (iλ-mode, where $i = 1, 2, 3$; $\lambda = 1, 2, 3, 4, 5, 6$) corresponding to all nonzero piezoelectric coefficients. To systematically explain the theoretical framework of the construction method for OSSPSU, Fig. 1 shows the relationship among the piezoelectric strain unit, the OSSPSU and the full-parameter artificial vibration mode through the synergistic strain effect of the topological structures. To this design, the orders of transverse vibration mode (showing extension or contraction deformation) and bending vibration mode are not limited, while Fig. 1 just shows a part of $x^{th}$ transverse or $y^{th}$ bending vibration modes. For the construction of the OSSPSU, we use piezoelectric strain units and arrange them orderly in three-dimensional (3D) space. Under the alternating current (AC) electric field, a large deformation can be generated in the OSSPSU along the preferred direction, and corresponding artificial vibration modes can be obtained through the synergistic strain effect between/among the piezoelectric strain units. The detailed design of OSSPSU construction can be categorized into OSSPSU that can induce normal strain and OSSPSU that can induce shear strain. Specifically, normal strain of OSSPSU can be achieved via transverse vibration mode, which is based on the shear-extension or normal-extension topological structures. On the other hand, shear strain of OSSPSU can be achieved via bending vibration mode, which is based on the shear-bending or normal-bending topological structures. In the expression of topological structures, the first word ("shear", "normal") corresponds to type of the strain of piezoelectric strain unit,

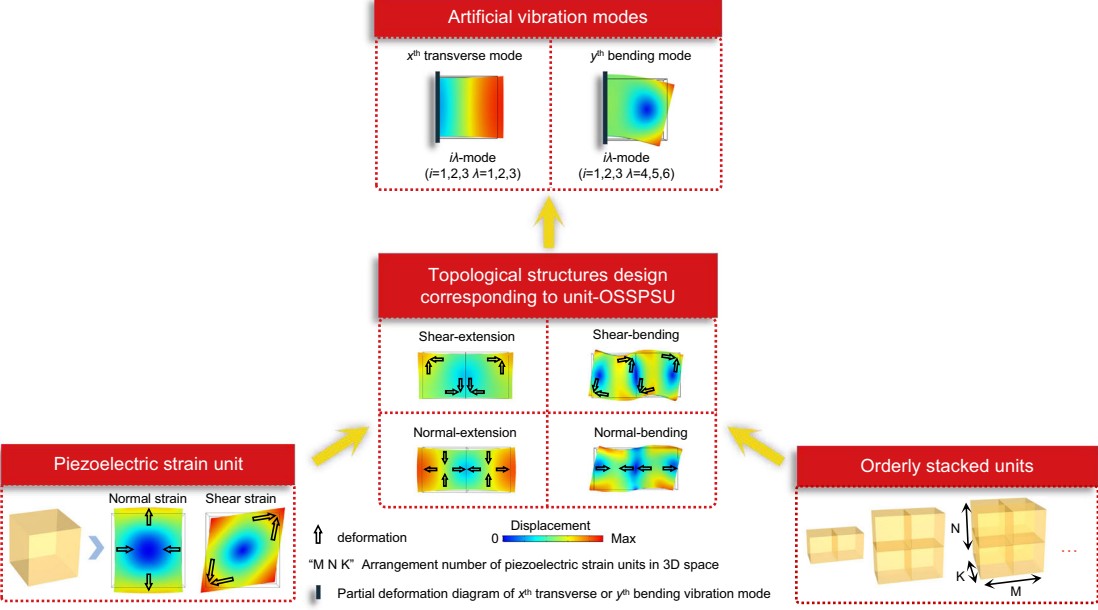

**Fig. 1 | Design concept of OSSPSU methodology.** The relationship among the single piezoelectric strain unit, orderly stacked units, topological structure design, and artificial vibration modes.

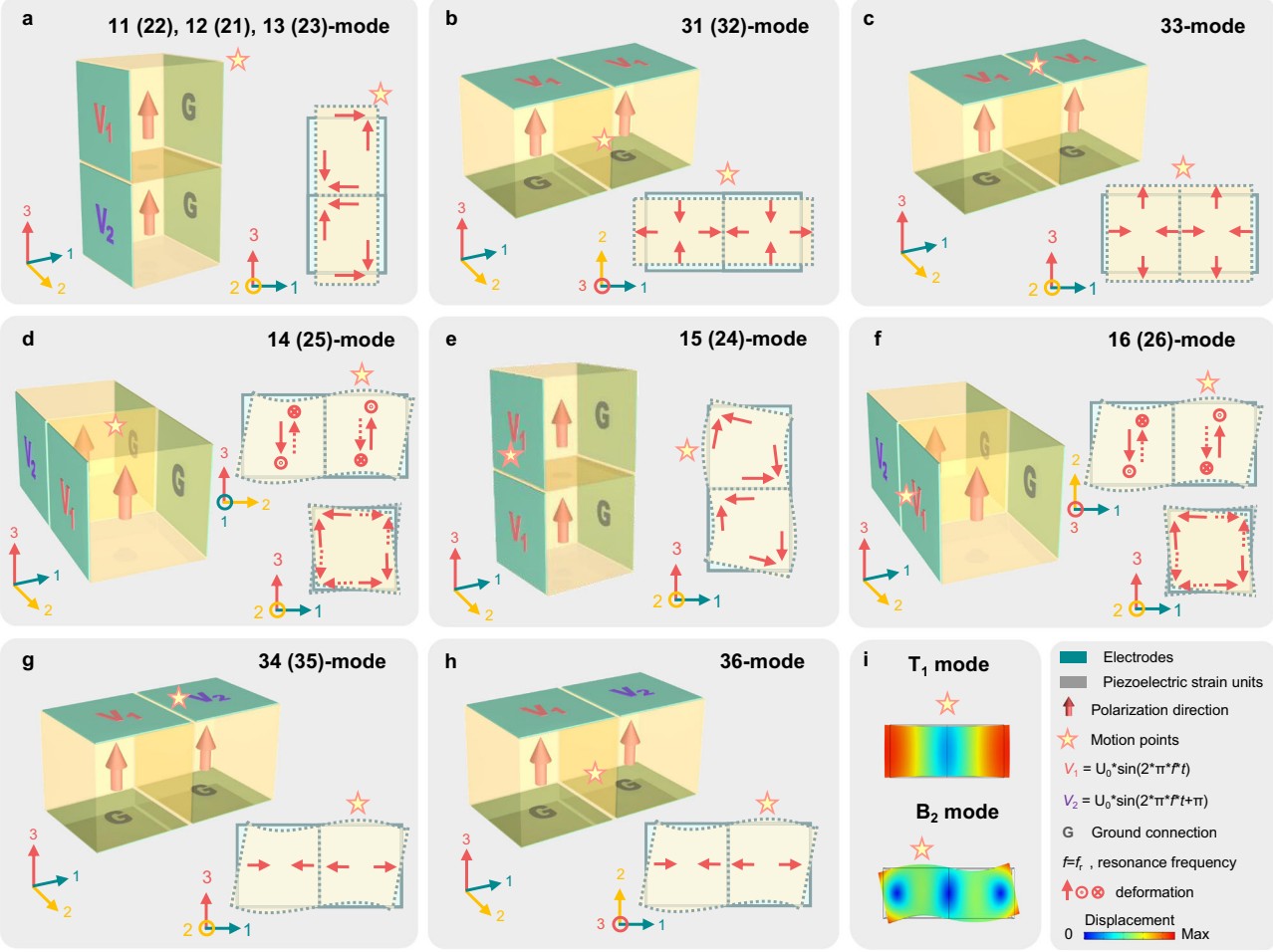

**Fig. 2 | Design of the two-unit OSSPSU. a** The applied electric voltage signals and deformation schematics of artificial 11 (22)-, 12 (21)- and 13 (31)- modes; **b** artificial 31 (32)-mode; **c** artificial 33-mode; **d** artificial 14 (25)-mode; **e** artificial 15 (24)-mode; **f** artificial 16 (26)-mode; **g** artificial 34 (35)-mode; **h** artificial 36-mode. **i** The simulated deformations diagrams of **a**–**h**.

and the second word ("extension", "bending") indicates the artificial vibration mode of OSSPSU.

In order to realize all the above artificial vibration modes and verify the generality of the method, an OSSPSU with two-unit is designed as an example. The methodology is verified by finite element simulation, theoretical analysis, piezoelectric stator measurement, and device demonstration in the following text.

Figure 2 is the schematic diagram of two-unit OSSPSU. Combined with the Finite Element Method (FEM) (COMSOL Multiphysics), the full artificial vibration modes can be excited. For all OSSPSUs in the figure, piezoelectric strain units are polarized in the positive direction along 3-axis, and all of the artificial vibration modes are excited under stress free boundary condition. To induce a normal strain (e.g., the first transverse ($T_1$) vibration mode) in OSSPSU, one can use the following two methods: (i) utilization of the 15- or 24-mode piezoelectric strain units to construct a shear-extension topological structure (Fig. 2a); (ii) utilization of the 31-, 32- or 33-mode piezoelectric strain units to construct a normal-extension topological structure (Fig. 2b, c). For shear-extension topological structure (as shown in Fig. 2a), the piezoelectric strain units are stacked along 3-axis, and the AC electric fields $V_1$, $V_2$ with phase difference of π are applied on the surface perpendicular to the 1-axis. In this case, the shear strains of the two piezoelectric strain units are opposite. As a result, the entirety of OSSPSU shows an integral extension or contraction deformation along 1-, 2-, or 3-axis, corresponding to artificial 11-, 12-, or 13-mode, respectively. Similarly, when $V_1$, $V_2$ are applied on the OSSPSU surface perpendicular to the 2-axis,

the 24-mode of piezoelectric strain units will be excited. This means that the OSSPSUs corresponding to artificial 22-, 21-, and 23-modes can also be constructed with shear-extension structure. In Fig. 2b, c, under the same voltage along 3-axis, the OSSPSUs with normal-extension topological structure show extension or contraction strain along 1-, 2-, and 3-axes, which correspond to artificial 31-, 32-, and 33-modes, respectively.

To induce a shear strain (e.g., the second bending ($B_2$) vibration mode) in OSSPSU, one can use the following two methods: (i) utilization of the 15- or 24-mode piezoelectric strain units to construct a shear-bending topological structure (Fig. 2d–f); (ii) utilization of the 31-, 32- or 33-mode piezoelectric strain units to construct a normal-bending topological structure (Fig. 2g, h). As shown in Fig. 2d, f, the piezoelectric strain units are stacked along 2-axis, and two AC electric fields with a phase difference of π are applied on the planes perpendicular to the 1-axis. Then, the two piezoelectric strain units of the OSSPSUs produces opposite shear strains ($\varepsilon_{13}$). Owing to synergistic strain effect, the OSSPSUs show an integral bending deformations in the plane perpendicular to 1-axis (Fig. 2d) and 3-axis (Fig. 2f), respectively, which are equivalent to artificial 14- and 16-modes. For OSSPSU representing the artificial 15-mode (Fig. 2e), two piezoelectric strain units are stacked along 3-axis. Under the bending vibration mode, an identical voltage applies on the surfaces of the units perpendicular to the 1-axis. Then, the artificial 15-mode of the OSSPSU will be excited and the distortion behavior will be observed in the plane perpendicular to 2-axis. Similarly, the OSSPSUs corresponding to artificial 25-, 24- and 26-modes can be

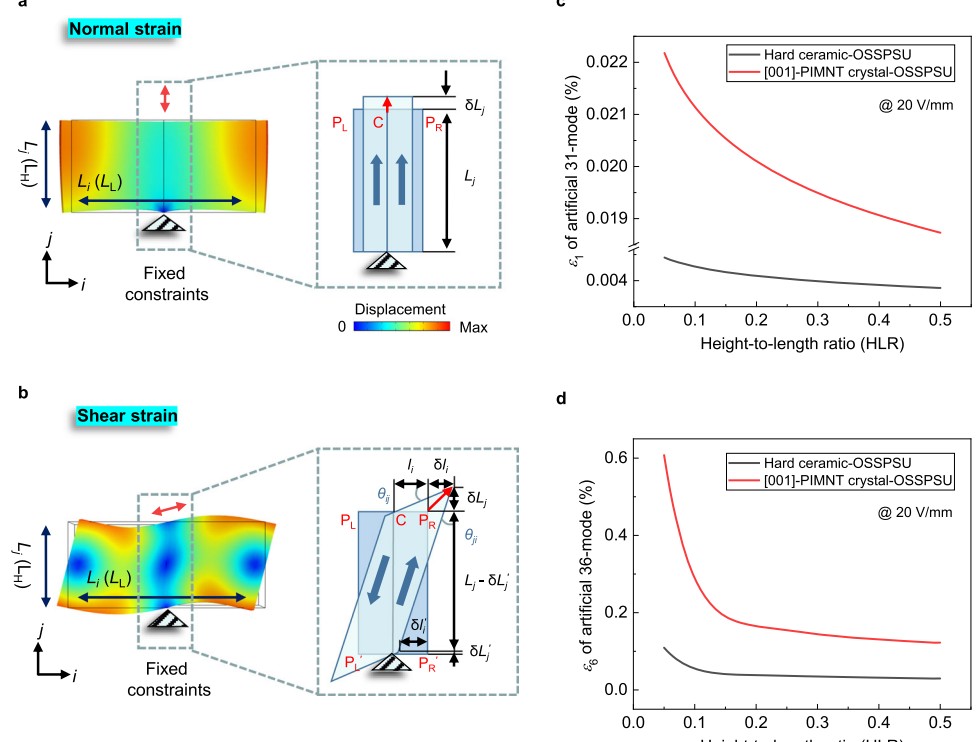

**Fig. 3 | Strains of artificial vibration modes with respect to geometric dimension of two OSSPSU calculated by FEM simulation. a** Geometrical diagram, boundary conditions, and deformations of the simulated normal strain ($\varepsilon_\lambda$: $\lambda$ = 1, 2, 3) by using OSSPSU; **b** shear strain ($\varepsilon_\lambda$: $\lambda$ = 4, 5, 6) by using OSSPSU. The black triangles represent the fixed constraint. The red double-headed arrows indicate the displacement directions of motion points $P_L$ (point on the left of the selected deformation region), C (point on the center of the selected deformation region), or $P_R$ (point on the right of the selected deformation region). A hard ceramic and a [001]-poled PIMNT single crystal[39,40] were chosen as demoed materials under the electric field of 20 V mm$^{-1}$. **c** The variation tendencies of strain of 31-mode with geometric dimension predicted by FEM simulation. **d** The variation tendencies of strain of 36-mode with geometric dimension predicted by FEM simulation.

constructed by using the units based on 24-mode. Using the normal piezoelectric strain units as basic piezoelectric strain unit, OSSPSUs with normal-bending topological structure corresponding to the artificial 34 (35)-mode (Fig. 2g), and 36-mode (Fig. 2h) can be constructed. In detail, two piezoelectric strain units are stacked along 1-axis, and AC electric fields $V_1$ and $V_2$ are applied on piezoelectric strain units along 3-axis, leading to extension and contraction deformation of each piezoelectric strain unit at the same time. Two piezoelectric strain units synergistically extrude one another, and shear deformations arise in the plane perpendicular to 1 (2)-axis, and 3-axis, which correspond to artificial 34 (35)-, and 36-modes. Figure 2i shows the deformation diagrams of $T_1$ and $B_2$ modes. Supplementary Movie 1 shows the dynamic deformation diagram of the simulated vibration mode for all the above OSSPSUs with artificial vibration modes.

Based on the same construction mechanism, by changing the ordered arrangement, dimensions or number of piezoelectric strain units, full artificial vibration modes can also be achieved, while the structural diagram of 2 × 2 arrayed OSSPSU is given in Supplementary Fig. 1. Significantly, due to the strain coupling and strong electromechanical coupling effect at the resonant state, the strain in some specific region of the OSSPSU may be reach an extremely high value.

**Output strains of OSSPSUs**
The strains ($\varepsilon_\lambda$s) corresponding to the artificial vibration modes of the OSSPSU are related to (i) intrinsic piezoelectric material properties, and (ii) topological structure and geometric dimensions. We adopted FEM simulation, theoretical analysis, and experimental results to investigate the strains of OSSPSUs with respect to the above two factors. Hard ceramics and [001]-poled PIMNT single crystals are chosen

to make piezoelectric stator. Simulation details can be found in Experimental Section.

As shown in Fig. 3a, b, the normal strain and shear strain can be induced by exciting the $T_1$ mode and $B_2$ mode of the OSSPSU, respectively. $L_j$ and $l_i$ are the outline dimension of the deformation region of the OSSPSU; $\delta L_j$ is the displacement along $j$-direction of the motion points $P_L$ (point on the left of the selected deformation region), C (point on the center of the selected deformation region), or $P_R$ (point on the right of the selected deformation region); $\delta l_i$ and $\delta l_i'$ are the displacements along $i$-direction of $P_R$ and $P_R'$, respectively; $\delta L_j'$ is the displacement along $j$-direction of $P_R'$; $\theta_{ij}$ and $\theta_{ji}$ are the shear angles in $i$-$j$ plane shown in Fig. 3b. The detailed mechanical analysis is presented in Supplementary Note 1.

As examples, Fig. 3c, d depict the variations of strains $\varepsilon_1$, $\varepsilon_6$ corresponding to artificial 31-, 36-modes of OSSPSUs as a function of height-to-length ratio (HLR) of the OSSPSU under the electric field of 20 V mm$^{-1}$, respectively. A detailed analysis for selected HLR is given in Supplementary Note 1. The results show that there is a nonlinear correlation between the strains and HLR. As the HLR increases, the $\varepsilon_1$ and $\varepsilon_6$ of the artificial 31- and 36-modes decrease and gradually approach a stable value. It indicates that the strain can be adjusted in a wide range by changing the HLR of the OSSPSU. In addition, it should be noted here that the strains of different vibration modes of a material are very different (Fig. 3c, d). It is clearly shown that strains of [001]-PIMNT crystal-OSSPSUs are about an order of magnitude higher than that of ceramic-OSSPSUs under the same vibration mode.

**Experimental investigation on OSSPSU stators**
To verify the feasibility of the OSSPSU, we fabricated a coupling mode OSSPSU stator and analyzed its properties. It is worth noting that the

working principle of the ultrasonic motors and other piezoelectric devices are the coupling of two orthogonal motion directions. Since the shear and normal strain can create the desired orthogonal motion, OSSPSUs can be combined in pairs, such as 11- and 15-modes OSSPSU pair, 33- and 34-modes OSSPSU pair, 31- and 36-modes OSSPSU pair. Specifically, here we demonstrate the 31−36 coupled vibration mode OSSPSU stator. To effectively excite the orthogonal motion of this 31−36 coupled mode OSSPSU stator, the resonant frequency of the 31 normal mode and the 36 shear mode should be as close as possible. Hence, the size of the crystal and ceramic OSSPSU stators are set to be 5 mm$^{Length}$ × 1.3 mm$^{Height}$ × 1.06 mm$^{Width}$ (HLR = 0.26) and 5 mm$^{Length}$ × 1.3 mm$^{Height}$ × 3.24 mm$^{Width}$ (HLR = 0.26), respectively. Detailed topological structure and geometrical dimension optimization is provided in Supplementary Fig. 2.

Figure 4a displays the schematic of the OSSPSU stator and Fig. 4b, c show impedance spectra of the [001]-PIMNT crystal and hard ceramic piezoelectric OSSPSU stators measured under stress free boundary condition. It can be seen that the resonant peaks of the two OSSPSU stators are split, indicating that 31- and 36-modes have been successfully coupled. In addition, Supplementary Note 2 analyzes the electromechanical coupling coefficient ($k_{eff}$) of the two OSSPSU stators, showing that the $k_{eff}$ of the crystal-OSSPSU stator is about twice that of the ceramic one, indicating that the former one is more efficient for electromechanical coupling. It should be noted here that piezoelectric stators used in electromechanical devices may be subjected to various external forces, including preload force, uniaxial stress or hydrostatic pressure, which may significantly affect the properties of piezoelectric stators. Supplementary Fig. 5 depicts the uniaxial stress dependence of impedance spectra of [001]-PIMNT crystal and hard ceramic piezoelectric OSSPSU stators. Detailed analysis can be found in Supplementary Note 4.

Table 1 lists the simulated and tested results of strains of 31-mode and 36-mode for OSSPSU stators made of [001]-PIMNT crystal and hard ceramic under the electric field of 20 V mm$^{-1}$. The simulated strains corresponding to 31−36 coupled vibration mode of the crystal-OSSPSU stator is around 0.039%. Nevertheless, as HLR = 0.26, the simulation strains of pure vibration modes of 31-mode and 36-mode are around 0.02% and 0.16% (inferred from Fig. 3),

respectively. The reason for the difference is that while the two modes are infinitely close to each other, the high-frequency electromechanical coupling effect will produce a certain weakening phenomenon, but it still remains at a high level. Moreover, the experimental strains ($\varepsilon_6$) of 31−36 coupled mode (0.026% for crystal-OSSPSU stator and 0.018% for ceramic-OSSPSU stator) were measured, and the detailed experiments and analysis are expounded in Supplementary Note 3. Obviously, both the simulated and tested strains ($\varepsilon_6$) of 31−36 coupled mode for crystal-OSSPSU stator are higher than that of ceramic one.

## The miniaturization linear OSSPSU motor with nanometer resolution

To experimentally verify practical application of the 31−36 coupled mode OSSPSU stator (5 mm$^{Length}$ × 1.3 mm$^{Height}$ × 1.06 mm$^{Width}$), we proposed a linear OSSPSU motor as shown in Fig. 5a. As far as we know, it is one of the smallest standing wave ultrasonic motors, which benefits from the adjustable size of the piezoelectric strain units of the OSSPSU. For a detailed theoretical derivation of the motion velocity of the OSSPSU motor is expounded in Supplementary Note 5. According to the inference results (Supplementary Eq. (S9)), the output speed of the OSSPSU motor is closely related to the $\varepsilon_6$ of 31−36 coupled mode and the preload force of the OSSPSU stator. To determine the optimal preload force of the OSSPSU stator, we designed a home-made test setup, as shown in Supplementary Fig. 6. By this test setup, the optimal preload force of PIMNT crystal-OSSPSU stator is found to be around 300 mN.

Figure 5b−e display the velocities of OSSPSU motors made of [001]-PIMNT crystal and hard ceramic with respect to the applied voltage and load. It can be seen that the motion speeds of the crystal-OSSPSU and ceramic-OSSPSU motors show a linear relationship with respect to the voltage and load. To ensure that the piezoelectric materials are not depolarized, in this work, the maximum AC driving voltage applied on the crystal-OSSPSU and ceramic-OSSPSU motors are set to be 150 $V_{P-P}$ and 250 $V_{P-P}$, respectively. Under the voltage of 150 $V_{P-P}$, the maximum speeds of the two channels of the crystal-OSSPSU motor are 30.4 ± 1.1 mm s$^{-1}$ and 32.1 ± 0.4 mm s$^{-1}$, respectively. As the load is 1.5 gram (g), the maximum speeds of the two channels are 23.5 ± 0.5 mm s$^{-1}$

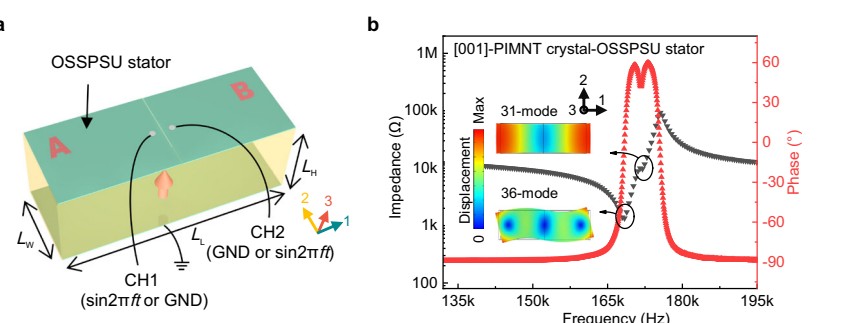

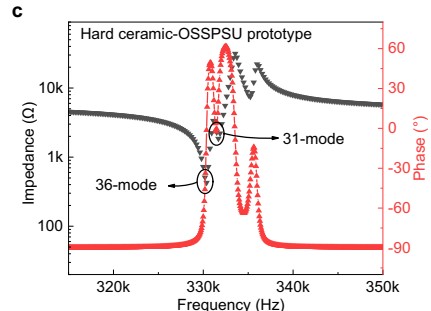

**Fig. 4 | Schematic, electric impedance and phase spectra of the 31−36 coupled mode OSSPSU stator. a** The stator schematic image. The red arrow shows the direction of poling. **b** The impedance and phase spectra of [001]-PIMNT crystal-OSSPSU stator. **c** The impedance and phase spectra of the hard ceramic-OSSPSU stators. The inset in **b** shows the schematic of vibration modes for 31 and 36 of the OSSPSU stators.

**Table 1 | The simulated and tested results of strains of artificial 31-mode ($\varepsilon_1$), 36-mode ($\varepsilon_6$), and 31–36 coupled mode ($\varepsilon_6$ for coupled mode) of OSSPSU stators based on [001]-PIMNT crystal and hard ceramic under the electric field of 20 V mm$^{-1}$**

| Piezoelectric stators | Specimen sizes (length [mm] × hight [mm] × width [mm]) | Simulation values for pure modes (%) | | Simulation values for coupled modes (%) | Experimental values for coupled modes (%) |
|---|---|---|---|---|---|
| | | $\varepsilon_1$ | $\varepsilon_6$ | $\varepsilon_6$ | $\varepsilon_6$ |
| [001]-PIMNT crystal-OSSPSU stator | 5 × 1.3 × 1.06 (HLR = 0.26) | 0.02 | 0.16 | 0.039 | 0.026 |
| Hard ceramic-OSSPSU stator | 5 × 1.3 × 3.24 (HLR = 0.26) | 0.004 | 0.04 | 0.028 | 0.018 |

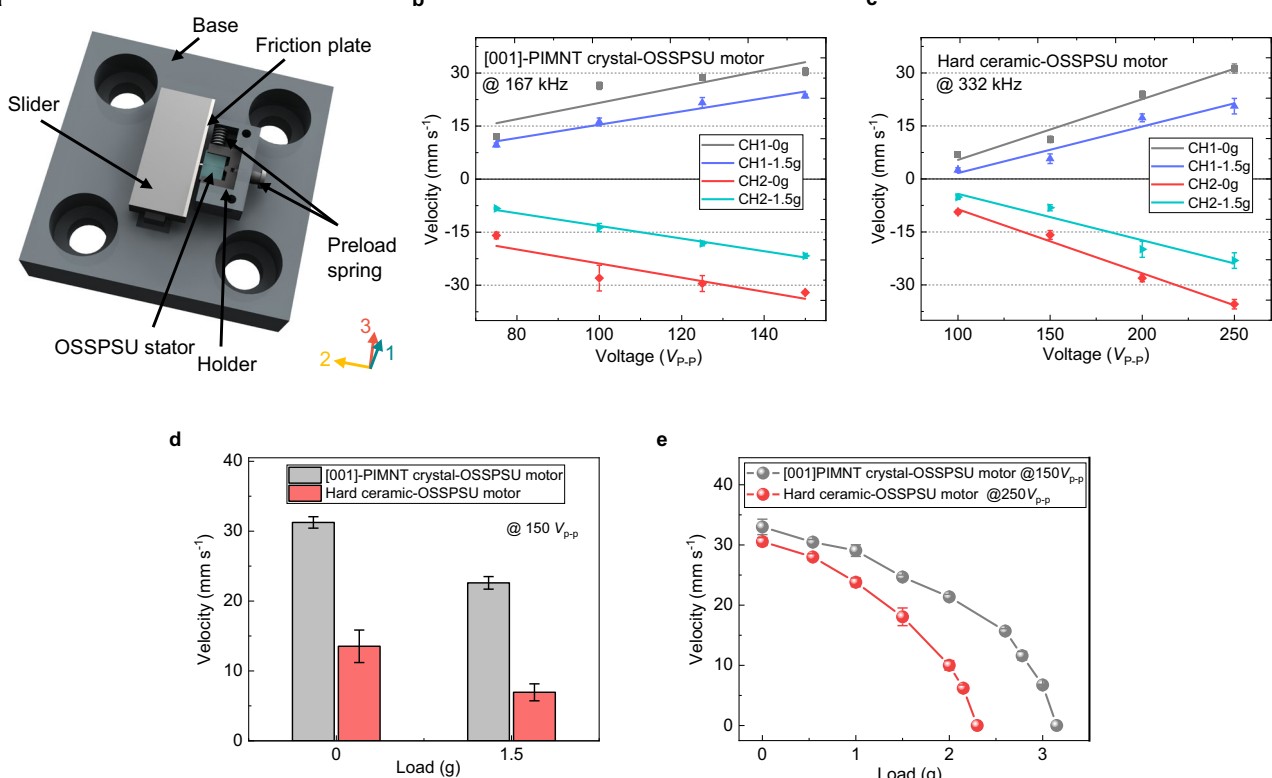

**Fig. 5 | Schematic of the proposed miniature linear OSSPSU motor and the velocity performance for the OSSPSU motors. a** Basic design of the miniature linear ultrasonic OSSPSU motor. **b** The relationship between the bidirectional motion velocities of [001]-PIMNT crystal-OSSPSU motor and the applied voltage and load. **c** The relationship between the bidirectional motion velocities of hard ceramic-OSSPSU motor and the applied voltage and load. **d** Comparison of the velocity performance for [001]-PIMNT crystal and hard ceramic-OSSPSU motors under the same conditions. **e** Plot of the velocities of [001]-PIMNT crystal-OSSPSU motor (at 150 $V_{P-P}$) and hard ceramic-OSSPSU motor (at 250 $V_{P-P}$) versus the load. The error bars in **b**, **c**, and **e** represent the standard deviations of the values of each motor measured three times under the same conditions.

and 21.7 ± 0.7 mm s$^{-1}$, respectively. As shown in Fig. 5d, under the same load and voltage, the velocity of the crystal-OSSPSU motor is more than two times that of the ceramic-OSSPSU motor. Figure 5e shows the curves of the motion speeds of crystal-OSSPSU motor (measured at 167 kHz and 150 $V_{P-P}$) and ceramic-OSSPSU motor (measured at 332 kHz and 250 $V_{P-P}$) with load. The experimental results show that the maximum thrust force of the crystal-OSSPSU motor is 3.15 g, which is about 1.4 times that of the ceramic-OSSPSU motor (2.3 g). Compared with the ceramic-OSSPSU motor, the higher motion velocity and thrust force of OSSPSU motor made from [001]-PIMNT crystals are attributed to the larger $\varepsilon_6$ of 31−36 coupled mode.

The graphical setup for measuring the resolution of the ultrasonic OSSPSU motors was established as shown in Fig. 6a. Figure 6b, c show the bidirectional step displacements as a function of voltage and time for OSSPSU motors under 30 cycles of a sinusoidal pulse signal. The step displacements reduce linearly with decreasing the voltage (as shown in Fig. 6d, e). Figure 6f, g show a good linear relationship between the step and the cycle of sinusoidal pulse signal for both [001]-PIMNT crystal and hard ceramic-OSSPSU motors. As shown in Fig. 6h, at resonant frequency the minimum step displacements of crystal-OSSPSU motor (measured at 150 $V_{P-P}$, 5 cycle) and ceramic-OSSPSU motor (measured at 250 $V_{P-P}$, 26 cycle) are 3 nm and 8 nm, respectively. Detailed experiments for the nanometer resolution can be found in Supplementary Note 6. The ultra-high resolution of the crystal-OSSPSU motor could be attributed to the miniaturization design of the OSSPSU stator. Furthermore, it is expected that higher displacement resolution can be obtained by utilizing a closed-loop control system[34,35].

Supplementary Table 2 summarizes the output performance of the nanostep piezoelectric motors in this work and previous reports. It can

be found that the OSSPSU motor based on the crystal-OSSPSU stator has the advantages of miniaturization design and nanoscale resolution. In addition, owing to the large $\varepsilon_6$ of 31−36 coupled mode, the crystal-OSSPSU motor possesses a large output force in unit volume of 4.5 mN mm$^{-3}$, and high speed in unit volume of 4.66 s$^{-1}$ mm$^{-2}$, which are 4.1 times and 3.1 times larger than that of the motor made of hard ceramic, respectively. To display the excellent output performance of the piezoelectric OSSPSU motor based on [001]-PIMNT single crystal-OSSPSU stator, an auto focus (AF) lens was designed as shown in Supplementary Note 7 and Movie 2. It is helpful to achieve rapid and precision lens focusing.

## Discussion

In summary, inspired by the concept of metamaterial design, we proposed an OSSPSU, which can excite full artificial vibration modes in the piezoelectric materials (such as piezoceramics), and/or ultrahigh strains, benefiting the design of new coupled piezoelectric vibration modes. In this work, an OSSPSU based on two [001]-poled PIMNT crystal piezoelectric strain units was fabricated with a 31−36 coupled mode. The strain $\varepsilon_6$ of 31−36 coupled mode was found up to 0.026% at the electric field of 20 V mm$^{-1}$. Based on the above OSSPSU stator, we design and fabricate a miniature linear piezoelectric motor (5 mm$^{Length}$ × 1.3 mm$^{Height}$ × 1.06 mm$^{Width}$), which is one of the smallest motors among all standing wave piezoelectric motors as far as we know. Owing to the ultrahigh $\varepsilon_6$ of 31−36 coupled mode of OSSPSU stator, the speed in unit volume of the crystal-OSSPSU motor is 4.66 s$^{-1}$ mm$^{-2}$, and its moving resolution is 3 nm, being two orders of magnitude higher than current ultrasonic piezoelectric motors. All these features indicate that our proposed

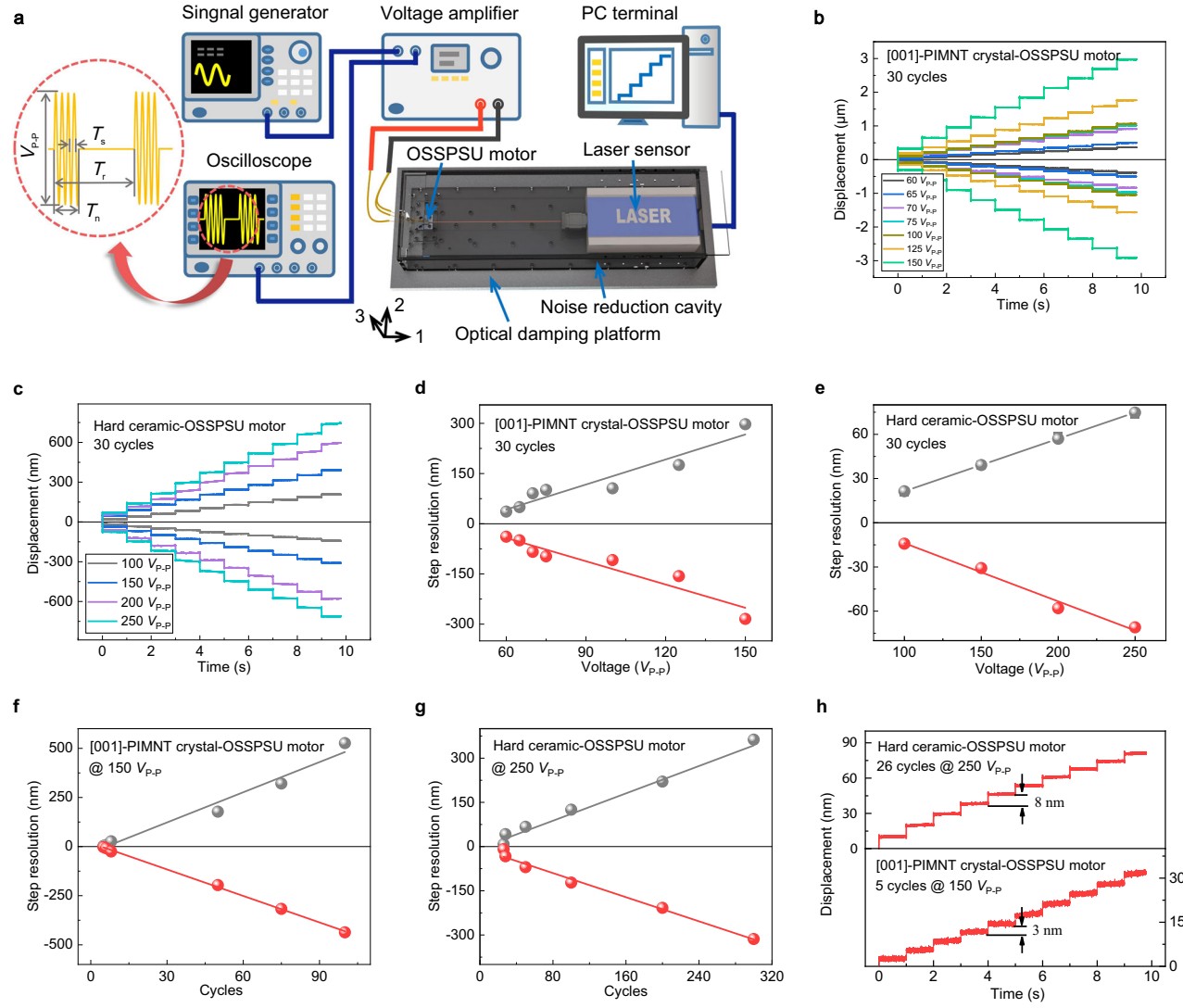

**Fig. 6 | Experimental results of nanostep motion characteristics for the OSSPSU motors under the same period of pulse signals. a** The graphical setup for measuring the resolution of the ultrasonic OSSPSU motors and the voltage driving signal for step motion. Where $T_s$ refers to the period of a single sinusoidal pulse signal, $T_n$ represents the period of n pulse signals, and $T_r$ (set to 1 second) represents the period of the step test. **b** The bidirectional motion steps of [001]-PIMNT crystal-OSSPSU motor under different voltages (at 30 cycle). **c** The bidirectional motion steps of hard ceramic-OSSPSU motor under different voltages (at 30 cycle). **d** The step displacements of [001]-PIMNT crystal -OSSPSU motor as a function of the applied voltage. **e** The step displacements of hard ceramic-OSSPSU as a function of the applied voltage. **f** The bidirectional motion displacement resolutions of [001]-PIMNT crystal-OSSPSU motor (at 150 $V_{P-P}$ and 167 kHz); **g** hard ceramic-OSSPSU motor (at 250 $V_{P-P}$ and 332 kHz). **h** The step displacements of [001]-PIMNT crystal and hard ceramic-OSSPSU motors as a function of the cycle of pulse signals.

OSSPSUs are promising to assist the design of advanced piezoelectric devices.

## Method
### FEM simulation of OSSPSU
We adopted commercial software (Comsol Multiphisics) for FEM simulation. The full matrix parameters of [001]-poled PIMNT single crystal for simulation were obtained from ref. 36, and the properties of hard ceramic were derived from the Comsol library. The geometrical diagrams of different OSSPSU and the applied electrical boundary conditions correspond to Fig. 2. The electric field intensity of 20 V mm⁻¹ is uniformly applied to simulate the transient displacement. In addition, the linear fixed constraint is set at the center position opposite to the strain direction of the OSSPSU for displacement measurement. The output data points are from the maximum deformation of the OSSPSU relative to the central position.

### Fabrication and measurement of OSSPSU stator
The material selected in this paper was the [001]-oriented rhombohedral (R) phase PIMNT single crystal grown by the modified Bridgman technique[37,38]. Then, cut the the crystal into a dimension of 5 mm$^{Length}$ × 1.3 mm$^{Height}$ × 1.06 mm$^{Width}$ along the direction of [100]/[010]/[001] in the pseudocubic coordinate system. The gold electrode with a thickness of about 100 nm was sputtered on the two surface ($L_L$ × $L_W$) of the crystal using a high vacuum ion sputtering instrument (Q150T S Plus, Quorum Technologies Limited Company, UK). Then used a dicing saw (DS616, China) to divide the upper electrode into two electrode surfaces evenly according to the direction of Fig. 4a, and three wires were bonded with room temperature silver paste. Then the crystal was polarized in silicon oil for 5 min under a 10 kV cm⁻¹ direct current electric field along the 3-axis at room temperature to form the OSSPSU stator. The hard ceramic-OSSPSU stator was prepared wtih the size of 5 mm$^{Length}$ × 1.3 mm$^{Height}$ × 3.24 mm$^{Width}$ by the same method.

The impedance magnitude and phase spectra characteristics of the [001]-PIMNT crystal and hard ceramic piezoelectric OSSPSU stators measured by using a LCR meter in the stress free boundary state (Agilent E4294A). We utilized standing wave method to drive the OSSPSU stator. As a sinusoidal AC voltage excitation signal ($\sin 2\pi f t$) with the frequency of resonant frequency was applied to channel 1 (CH1), meanwhile, channel 2 (CH2) was for grounding, 31 and 36 modes can be excited simultaneously. The coupling mode further drove the OSSPSU to form a continuous microscopic oblique linear motion (as shown in Supplementary Fig. 3a). Likewise, switching the applied voltage mothed of CH1 and CH2, the OSSPSU can produce continuous oblique liner motion in the opposite direction (as shown in Supplementary Fig. 3b). Supplementary Fig. 12 shows the experimental setup for the frequency dependence of the displacements of the OSSPSU stator. The stator sample is fixed on the optical damping platform. The sinusoidal wave generated by the signal generator (DG1032Z, Rigol Technologies, China) was applied to the piezoelectric OSSPSU through the voltage amplifier (ATA-4052, Agitek, China). Then the displacements of the OSSPSU stator were measured using the high-precision laser sensors (LK-G30, Keyence).

### Fabrication and measurement of the OSSPSU motor

The $ZrO_2$ friction tip with the size of φ0.8 mm ($l$, diameter) × 0.5 mm ($h$, height) was bonded to the center of the surface ($L_L \times L_H$) with epoxy (Catalyst 15 Black, Emerson and Cuming, Germantown, WI, USA), leading to greatly reduced size of the OSSPSU stator compared with the traditional $L_1B_2$ piezoelectric stator. Figure 5a shows the schematic of the proposed OSSPSU motor, which consists of a base, an OSSPSU stator, a stator holder, a slider with a friction plate made of zirconia, and a preloading structure composed of springs and screw.

The experimental setup is shown in Supplementary Fig. 12. The stator sample is fixed on the optical damping platform. The sinusoidal wave generated by the signal generator (DG1032Z, Rigol Technologies, China) was applied to the piezoelectric OSSPSU stator through the voltage amplifier (ATA-4052, Agitek, China). Then the output characteristics were measured using the high-precision laser sensors (LK-G30, Keyence for velocity measurement; and LV-S01, Sunnyor for step measurement). The graphical setup for measuring the resolution of the ultrasonic OSSPSU motors was established, as shown in Fig. 6a. The OSSPSU motor is fixed in the noise reduction cavity and then fasted on the optical damping platform together. Figure 6a displays the voltage driving signal for step motion, where $T_s$ refers to the period of a single sinusoidal pulse signal, $T_n$ represents the period of n pulse signals, and $T_r$ (set to 1 s) represents the period of the step test. Figure 6b–e show the bidirectional step displacements as a function of voltage and time for OSSPSU motors under 30 cycles of a sinusoidal pulse signal. These data are measured under open-loop control condition. The step displacements reduce linearly with decreasing the voltage (as shown in Fig. 6d, e). The mechanical loading is tested with the string pulley-weight system.

### Reporting summary

Further information on research design is available in the Nature Research Reporting Summary linked to this article.

## Data availability

The data that support the findings of this study are included with the manuscript as Supplementary Information. Any other relevant data are also available upon request from X.G. and F.L.

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

## Acknowledgements

This work was supported by the National Natural Science Foundation of China (Grant nos. 52002312, 51922083, 51831010, and 52172129), Natural Science Foundation of Shaanxi Province (Grant nos. 2021GXLH-Z-025 and 2020JM-004), and OPPO Research Fund.

## Author contributions

The work was conceived and designed by J.L., X.G., and F.L.; J.L. performed the finite element simulation and fabricated the samples and performed the experiments; J.L. and X.G. performed the electro-mechanical properties measurement; H.J. and K.R. assisted the fabrication of the stators; L.Q. and C.Q. assisted the properties measurements for the stators; J.G. assisted the design of the motor fixture; W.C. and Y.H. assisted the manufacture of motor fixtures; F.L., X.G., and Z.X. supervised the fabrication and test of the stators; J.L., X.G., and F.L. drafted the manuscript; S.D. revised the manuscript; and all authors discussed the results.

## Competing interests

The authors declare no competing interests.

## Additional information

Jinfeng Liu[1], Xiangyu Gao ⓘ [1] ✉, Haonan Jin[1], Kaile Ren[1], Jingyu Guo[1], Liao Qiao[1], Chaorui Qiu[1], Wei Chen[2], Yuhang He[2], Shuxiang Dong[3,4], Zhuo Xu[1] & Fei Li ⓘ [1] ✉

[1]Electronic Materials Research Laboratory, Key Lab of Education Ministry and State Key Laboratory for Mechanical Behavior of Materials, School of Electronic Science and Engineering, Xi'an Jiaotong University, Xi'an, China. [2]OPPO Guangdong Mobile Communication Co., Ltd., Shenzhen, China. [3]School of Materials Science and Engineering, Peking University, Beijing, China. [4]Institute for Advanced Study, Shenzhen University, Shenzhen, China. ✉e-mail: gaoxiangyu@xjtu.edu.cn; ful5@xjtu.edu.cn

