## [Peer Review File · Nature Communications]

Miniaturized electromechanical devices with multi-vibration modes achieved by orderly stacked structure with piezoelectric strain unitsREVIEWER COMMENTS

Reviewer #1 (Remarks to the Author):

An OSSPSU was proposed in this paper, which can excite full artificial vibration modes in the piezoelectric materials, such as piezoceramics, and/or ultrahigh strains, benefiting the design of new coupled piezoelectric vibration modes. Based on the OSSPSU prototype, a miniature linear piezoelectric motor was designed and fabricated, which realize excellent performance.

The paper has completed a good research on the metamaterial design, especially in the piezoelectric material. The paper could be considered publication in Nature Communication. However, the following points should be addressed by the authors.

- 1 Whether the properties are affected by external forces, such as preload.
- 2 Will the motions in different modes be coupled and cause interference?
- 3 What is the electrode material and whether it will affect the movement of the piezoelectric material or the installation in the motor?
- 4 Is there any performance difference between materials in different modes?

Reviewer #2 (Remarks to the Author):

The paper written by Liu J. et al discussed an orderly stacked structure with piezoelectric strain units (OSSPSU). By this method, the authors achieved artificial vibration modes and artificially coupled multi-vibration modes with ultra-high strains, which broke through the design bottleneck of the piezoelectric devices. Furthermore, the authors successfully fabricated a miniaturized piezoelectric motor based on a 31-36 coupled mode OSSPSU. Specifically, the newly reported motor showed great improvements in both speed and precision. This work is quite interesting to me and contains new design strategy, piezoelectric devices and results. In my opinion, this manuscript is suitable to publish in Nature Communications as the authors addressed the following comments.

- 1) For the artificial piezoelectric vibration modes (Fig. 1), what is the meaning of the deformation diagrams? The authors have to give corresponding explanations in the text or figures.
- 2) In Fig. 2, the authors describe the construction method of OSSPSUs to obtain all artificial vibration modes. What are the boundary conditions for using this method?
- 3) According to Fig. 3, the strain of the pure artificial 36-mode is much higher than that of the 31-36 coupled vibration mode. Why didn't use the pure 36-mode to design the piezoelectric motor? What's the merit(s) for the present design?
- 4) In the experimental investigation on OSSPSU prototypes on page 11, why did the authors choose the OSSPSU with HLR=0.26 to make piezoelectric motor? Is this the optimal dimensions?
- 5) In the miniaturization linear OSSPSU motor with nanometer resolution on page 12, why was the voltage applied on the motor made of PIN-PMN-PT crystals much lower than that applied on the ceramic motor? If the applied voltage is enlarged for the crystal motor, can

the motor exhibits even higher output performance (speed or resolution)?

Reviewer #3 (Remarks to the Author):

The authors present a comprehensive work on design strategy of metamaterial with non-zero piezoelectric coefficients with supporting FEM simulations.

The reviewer is however unable to appreciate the novelty of this work given similar earlier publications like <https://www.science.org/doi/full/10.1126/sciadv.aax1782>

Hence, I find it not suitable for publication in Nature communications.

Response Letter to Reviewers' Comments

Dear Editor and Referees:

We would sincerely thank the referees for their time and effort in carefully reading the manuscript and in preparing the review reports. We truly appreciate their positive comments and valuable suggestions. We have revised our manuscript accordingly, and as a result we believe its quality is greatly improved. The point-by-point responses to comments are enclosed. We hope we have satisfactorily addressed all referees' concerns and questions.

[Referees' comments are in black; Author responses are in blue; Revisions in the manuscript are highlighted.]

Response to reviewer #1

Comment: An OSSPSU was proposed in this paper, which can excite full artificial vibration modes in the piezoelectric materials, such as piezoceramics, and/or ultrahigh strains, benefiting the design of new coupled piezoelectric vibration modes. Based on the OSSPSU prototype, a miniature linear piezoelectric motor was designed and fabricated, which realize excellent performance.

The paper has completed a good research on the metamaterial design, especially in the piezoelectric material. The paper could be considered publication in Nature Communication. However, the following points should be addressed by the authors.

Reply: We appreciate the referee's positive comments and recommendation very much.

Comment: 1) Whether the properties are affected by external forces, such as preload.

Reply: We thank the reviewer for the valuable comment. In the following, we would like to discuss the influence of external force on the properties of piezoelectric material, orderly stacked structure with piezoelectric strain units (OSSPSU) and the piezoelectric motor, respectively.

(1) Effect of external force on the properties of piezoelectric materials

The effect of external force on the properties of piezoelectric material is closely related to material composition, ferroelectric phase and the magnitude and direction of applied force. Here, we give an example for the effect of uniaxial stress on the properties of relaxor-PT crystals.

For [001]-poled rhombohedral (R) PMNT crystal, the piezoelectric coefficient d_{33} and electromechanical coupling factor k_{33} as a function of uniaxial stress (σ_{33}) are shown in Fig. R1a [Zhang, S. *et al. J. Appl. Phys.*, 111, 2 (2012)]. The d_{33} was found to increase from 1700 to 2500 pC/N with increasing stress, while the k_{33} showed a slight decrease from 0.95 to 0.90. The increase in piezoelectric coefficient under uniaxial stress can be attributed to the fact that the stress σ_{33} moves the R-O phase boundary to a lower PT region, i.e., the R phase crystal approaches the R-O phase boundary under stress and give rise to improved piezoelectric properties. Figure R1b [Zhang, S. *et al. J. Appl. Phys.*, 111, 2 (2012)] shows the strain behaviors for [001]-poled R PMNT crystal under various levels of uniaxial stress. It can be observed that the value of strain increased from 2.7×10^{-3} to 4.2×10^{-3} under a stress from 0 to 40 MPa, owing to the stress induced the movement of

R-O phase transition. If the applied uniaxial stress is high enough (≥ 60 MPa), the piezoelectric response of [001]-poled PMNT crystals will decrease due to the stress induced depolarization of the crystals.

Fig. R1 The uniaxial stress dependence of piezoelectric properties of [001]-poled rhombohedral (R) phase PMNT crystals. a, Piezoelectric coefficient d_{33} and electromechanical coupling factor k_{33} behaviors for PMNT crystals under uniaxial stress σ_{33} . **b,** Strain-electric field behaviors for PMNT crystals under uniaxial stress σ_{33} . [Zhang, S. *et al. J. Appl. Phys.*, 111, 2 (2012)]

(2) Effect of external force on the properties of OSSPSU stator

We investigated the effect of uniaxial stress on the impedance spectra of 31-36 coupled mode OSSPSU stators by a home-made uniaxial stress loading setup [Xu, Y. *et al. J. Am. Ceram. Soc.*, 94, 3863 (2011)]. The stress dependence of impedance spectra of OSSPSU stators were measured by the LCR meter (Agilent E4294A). During the test, the uniaxial stress was applied perpendicular to the polarization direction, as shown in Fig. R2a. Figures R2b-c show the test results of the [001]-PIMNT crystal and hard ceramic piezoelectric OSSPSU stators. It can be seen that, with increasing the uniaxial stress, the intensity of both resonance and anti-resonance peaks in impedance curves decrease, which become more diffuse and shift towards higher frequencies.

Furthermore, with the increase of uniaxial stress, the impedance spectra of the [001]-PIMNT crystal and hard ceramic piezoelectric OSSPSU stators tend to split. This phenomenon shows that the OSSPSU stator is possible to be decoupled. Therefore, for electromechanical devices under the working condition with large uniaxial stresses, the OSSPSU should be carefully designed to avoid the decoupling of vibration modes.

Fig. R2 The schematic figure of OSSPSU stator, and the uniaxial stress dependence of impedance spectra of 31-36 coupled mode OSSPSU stators. **a**, The schematic figure of OSSPSU stator (red arrows represent polarization direction, and blue arrows represent the applied uniaxial stress). **b**, The impedance of [001]-PIMNT crystal-OSSPSU stator measured under uniaxial stress. **c**, The impedance of the hard ceramic-OSSPSU stator measured under uniaxial stress. (This figure has been added as Supplementary Fig. 5 in the revised manuscript.)

(3) Effect of external force on the performance of piezoelectric motor

The output performance of a piezoelectric motor is highly related to the external force, such as preload. There is an optimal preload to make the motor show maximum output performance. Detailed formulas are given in the Supplementary Information Section 4 of the revised manuscript, where the v_s (the motion velocity of the slider) can be calculated as follows:

$$v_s = \int \frac{(\mu_p + \mu_s)[F_p + k_e \theta_{12}(l/2)\sin(2\pi ft)] + k_e \theta_{21} L_w \sin(2\pi ft) + \mu_s Mg}{M} dt \quad (R1)$$

where F_p is the applied preload force. To determine to the optimal preload force of the OSSPSU stator, we designed a home-made test setup, as shown in Fig. R3. By this test setup, the optimal preload force of PIMNT crystal-OSSPSU stator is found to be about 300 mN.

Fig. R3 Home-made preload force test setup of the OSSPSU stator. (This figure has been added as Supplementary Fig. 6 in the revised manuscript.)

According to this comment, we added the following description in the revised manuscript:

“It should be noted here that piezoelectric stators used in electromechanical devices may be subjected to various external forces, including preload force, uniaxial stress or hydrostatic pressure, which may significantly affect the properties of piezoelectric stators. Supplementary Fig. 5 depicts the uniaxial stress dependence of impedance spectra of [001]-PIMNT crystal and hard ceramic piezoelectric OSSPSU stators. Detailed analysis can be found in Supplementary Section 4.”

“To determine the optimal preload force of the OSSPSU stator, we designed a home-made test setup, as shown in Supplementary Fig. 6. By this test setup, the optimal preload force of PIMNT crystal-OSSPSU stator is found to be around 300 mN.”

Comment: 2) Will the motions in different modes be coupled and cause interference?

Reply: Thanks for the valuable question. Yes, it is possible that the motions in different modes are coupled and cause interference. Actually, in practical applications of piezoelectric motors, we have to design specific coupled mode, such as 31-36 coupled mode, to achieve the continuous microscopic oblique linear or elliptical motion of the piezoelectric stator. Based on the coupled mode, the piezoelectric stator pushes the slider or rotor to perform macroscopic linear or rotational motion through friction. Furthermore, we have to avoid the interference from unwanted vibration modes. In our present work, the realization of the required coupled modes and the avoidance of unwanted modes for a piezoelectric stator was achieved by optimizing the geometry and electrodes of piezoelectric metamaterials.

Comment: 3) What is the electrode material and whether it will affect the movement of the piezoelectric material or the installation in the motor?

Reply: We thank the reviewer for the question. In this work, the electrode material is gold. Specifically, we sputter the gold electrodes with a thickness of about 100 nm on piezoelectric materials through a high vacuum ion sputtering instrument (Q150T S Plus, Quorum Technologies Limited Company, UK). The electrodes are very thin and light when compared to the piezoelectric stator. Thus, we didn't find any obvious impact of the electrode materials on the movement of the piezoelectric material and the installation in the motor.

According to this comment, we added the following description in the Method section of the revised manuscript:

“The gold electrode with a thickness of about 100 nm was sputtered on the two surface ($L_L \times L_W$) of the crystal using a high vacuum ion sputtering instrument (Q150T S Plus, Quorum Technologies Limited Company, UK).”

Comment: 4) Is there any performance difference between materials in different modes?

Reply: We thank the reviewer for this question. Yes, the performance of a material under different modes are different. For [001]-PIMNT crystal OSSPSU, the 31-mode and 36-mode can be realized

by exciting the first transverse (T_1) vibration mode and second bending (B_2) vibration mode of the OSSPSU, respectively. As shown in Fig. R4, the ε_6 corresponding to artificial 36-mode is much higher than the ε_1 corresponding to artificial 31-mode at the same HLR.

Table R1 listed the simulated strains corresponding to 31-mode, 36-mode and 31-36 coupled mode when HLR=0.26, which are 0.02%, 0.16% and 0.039%, respectively. This result indicates that the material shows different performances under different vibration modes.

According to this comment, we added the following description in the revised manuscript:

“In addition, it should be noted here that the strains of different vibration modes of a material are very different (Figs. 3c-d).”

Fig. R4 Strains of artificial modes with respect to geometric dimension of [001]-PIMNT crystal-OSSPSU stator calculated by FEM simulation under the electric field of 20 V mm⁻¹. **a**, The variation of the strain ε_1 of 31-mode with geometric dimension at resonance state; **b**, The variation of the strain ε_6 of 36-mode with geometric dimension predicted by FEM simulation at resonance state.

Table R1. The simulated results of strains of artificial 31-mode (ε_1), 36-mode (ε_6), and 31-36 coupled mode (ε_6 for coupled mode) of [001]-PIMNT crystal-OSSPSU stator under the electric field of 20 V mm⁻¹.

Piezoelectric stators	Specimen sizes (length [mm] × height [mm] × width [mm])	Simulation values for pure modes (%)		Simulation values for coupled modes (%)
		ε_1	ε_6	ε_6
[001]-PIMNT crystal-OSSPSU stator	5 × 1.3 × 1.06 (HLR=0.26)	0.02	0.16	0.039

Response to reviewer #2

Comment: The paper written by Liu J. *et al.* discussed an orderly stacked structure with piezoelectric strain units (OSSPSU). By this method, the authors achieved artificial vibration modes and artificially coupled multi-vibration modes with ultra-high strains, which broke through the design bottleneck of the piezoelectric devices. Furthermore, the authors successfully fabricated a miniaturized piezoelectric motor based on a 31-36 coupled mode OSSPSU. Specifically, the newly reported motor showed great improvements in both speed and precision. This work is quite interesting to me and contains new design strategy, piezoelectric devices and results. In my opinion, this manuscript is suitable to publish in Nature Communications as the authors addressed the following comments.

Reply: We thank the reviewer for his or her appreciation for the importance of our work.

Comment: 1) For the artificial piezoelectric vibration modes (Fig. 1), what is the meaning of the deformation diagrams? The authors have to give corresponding explanations in the text or figures.

Reply: This is a very good suggestion. According to the referee's comments, as shown in Fig. R5, we changed "Transverse mode" and "Bending mode" in the artificial piezoelectric vibration modes part of Fig. 1 to " x^{th} transverse mode" and " y^{th} bending mode", respectively. And we added the following description of the black symbol in the artificial piezoelectric vibration modes part of Fig. 1:

"Partial deformation diagram of x^{th} transverse or y^{th} bending vibration mode"

Accordingly, we updated Fig. 1 as Fig.R5 in the revised manuscript. In addition, we added the following description for the artificial piezoelectric vibration modes about Fig. 1 in the revised manuscript:

"To this design, the orders of transverse vibration mode (showing extension or contraction deformation) and bending vibration mode are not limited, while Fig. 1 just shows a part of x^{th} transverse or y^{th} bending vibration modes."

Fig. R5 Design concept of OSSPSU methodology. The relationship among the single piezoelectric strain unit, the OSSPSU and the artificial piezoelectric vibration mode via the synergistic strain effect of the topological structures at resonance modes. (Fig. 1 in the revised manuscript has been updated.)

Comment: 2) In Fig. 2, the authors describe the construction method of OSSPSUs to obtain all artificial vibration modes. What are the boundary conditions for using this method?

Reply: We thank the reviewer for this comment. In our work, all of the artificial vibration modes are excited under free boundary condition. According to this comment, we added the following statements in the revised manuscript:

“For all OSSPSUs in the figure, piezoelectric strain units are polarized in the positive direction along 3-axis, and all of the artificial vibration modes are excited under stress free boundary condition.”

Fig. R6 Design of the two-unit OSSPSU. **a**, The applied electric voltage signals and deformation schematics of artificial 11 (22)-, 12 (21)- and 13 (31)- modes; **b**, artificial 31 (32)- mode; **c**, artificial 33-mode; **d**, artificial 14 (25)-mode; **e**, artificial 15 (24)-mode; **f**, artificial 16 (26)-mode; **g**, artificial 34 (35)-mode; **h**, artificial 36-mode at resonance frequencies. **i**, The simulated deformations diagrams of **a-h**. (Fig. 2 in the revised manuscript has been updated.)

Comment: 3) According to Fig. 3, the strain of the pure artificial 36-mode is much higher than that of the 31-36 coupled vibration mode. Why didn't use the pure 36-mode to design the piezoelectric motor? What's the merit(s) for the present design?

Reply: We appreciate the very valuable comment. We agree with the referee that the pure artificial 36-mode is much higher than that of the 31-36 coupled vibration mode. Especially, when HLR of the OSSPSU is less than 0.1, the strain of pure artificial 36-mode increases approximately exponentially (Fig. R4b). However, for the following reasons, we choose the 31-36 coupled mode instead of pure 36 mode.

(1) If the HLR of piezoelectric materials is too small, the risk of cracks will be greatly increased due to stress concentration during the working process, so that the device has the risk of failure [Li, X. *et al. IEEE. T. Ultrason. Ferr*, 58(4), 698-703 (2011)]. Therefore, the performance is not the only criterion for device design.

(2) In principle, most ultrasonic motors require the coupling of two modes. For the ultrasonic motors in the manuscript, the motion directions of the two modes need to be mutually orthogonal. By combining the mode design and the input electric field design, the two orthogonal motions can

be coupled into an elliptic or oblique linear motion, and then the slider can be pushed to carry out linear or rotational motion. Therefore, the piezoelectric devices need to be optimized by FEM simulation to design the most reasonable size so as to realize the mode coupled.

As discussed above, the performance is not the only criterion for device design. Regarding the last part of the reviewer's comment (i. e., what's the merit(s) for the present design?), the present design enables the merits of compact size, coupled vibrational mode, and high step resolution. Specifically, the demonstrated motor is one of the smallest ultrasonic motors reported so far. Due to the adjustable size of the piezoelectric strain units, it is expected to design smaller piezoelectric motors. Owing to the high strain of the 31-36 coupled vibration mode, the speed in unit volume of the OSSPSU motor is up to $4.66 \text{ s}^{-1} \text{ mm}^{-2}$, and its moving resolution is $\sim 3 \text{ nm}$, being two orders of magnitude higher than that of state-of-the-art ultrasonic piezoelectric motors. These characteristics indicate that our proposed OSSPSU is helpful for the design of advanced miniaturized piezoelectric devices and is expected to be widely used in the fields of precision manufacturing, precision positioning and M/NEMS.

Comment: 4) In the experimental investigation on OSSPSU prototypes on page 11, why did the authors choose the OSSPSU with HLR=0.26 to make piezoelectric motor? Is this the optimal dimensions?

Reply: Thanks for the comment. HLR=0.26 is the optimal dimension in our work. To excite the 31-36 coupled mode of the piezoelectric motor, we use FEM simulation to optimize the dimension of the OSSPSU. Furthermore, to effectively excite the motion of the 31-36 coupled mode OSSPSU, the resonance frequencies of the 31 normal mode and the 36 shear mode should be as close as possible. The detailed FEM simulation process is shown in Fig. R7. Hence, the size of the crystal-OSSPSU stator in our work are set to be $5 \text{ mm}^{\text{Length}} \times 1.3 \text{ mm}^{\text{Height}} \times 1.06 \text{ mm}^{\text{Width}}$ (HLR=0.26) to make the resonance frequencies of 31 and 36 modes overlapped.

Fig. R7 Variation of frequencies of the 31- and 36-modes with L_w (width) for [001]-PIMNT single crystal-OSSPSU stator calculated by FEM simulation.

Comment: 5) In the miniaturization linear OSSPSU motor with nanometer resolution on page 12, why was the voltage applied on the motor made of PIN-PMN-PT crystals much lower than that

applied on the ceramic motor? If the applied voltage is enlarged for the crystal motor, can the motor exhibits even higher output performance (speed or resolution)?

Reply: Thanks for this valuable comment. The reason that the applied voltage on PIN-PMN-PT crystal is lower than that on the PZT4 ceramic is due to the fact that the coercive field of PIN-PMN-PT crystal is smaller than that of hard ceramic. Please check below for the detailed explanation of this comment.

(1) The allowable AC driving electric field of the piezoelectric devices should be less than half of the coercive fields of materials, otherwise there is a risk of depolarization.

From previous investigations, the allowable AC electric fields of samples for relaxor-PT crystals were less than half of their coercive fields, limiting the output strain [Zhang, S. et al. IEEE. T. Ultrason. Ferr, 58(2), 274-280 (2011)]. As reported in reference by Zhang, S. et al. Prog. Mater. Sci., 68, 1-66 (2015), the ECs for [001]-PIMNT crystal and hard ceramic are around 5 kV cm⁻¹ and 15 kV cm⁻¹, respectively. Therefore, the AC driving electric field of the [001]-PIMNT crystal should not exceed 2.5 kV cm⁻¹, and that of hard ceramics should not exceed 7.5 kV cm⁻¹.

(2) In the case that the material is not depolarized, increasing the AC electric field can make the motor exhibit higher output speed but lower resolution.

Regarding the last question of the reviewer's comment, the output performance of the motor, such as speed and mechanical load capacity, can be improved by appropriately increasing the voltage. Figures R8a-b display the velocities of OSSPSU motors made of [001]-PIMNT crystal and hard ceramic with respect to the applied voltage and load. It can be seen that the motion speeds of the crystal-OSSPSU and ceramic-OSSPSU motors show a linear relationship with respect to the voltage and load. Under the voltage of 150 V_{P-P} , the maximum speeds of the two channels of the crystal-OSSPSU motor are around 30 mm s⁻¹ and 32 mm s⁻¹, respectively. As the load is 1.5 gram (g), the maximum speeds of the two channels are around 23 mm s⁻¹ and 22 mm s⁻¹, respectively. According to the above results, it can be inferred that the output speeds of the motors can be increased when the driving electric field is appropriately increased.

Figures R8c-d show that the step displacements of both [001]-PIMNT crystal and hard ceramic-OSSPSU motors increase linearly with the increase of applied voltage when the cycle of sinusoidal pulse signal is constant. Theoretically, the smaller step displacement of the piezoelectric device corresponds to the higher positioning resolution. Therefore, in this case, increasing the applied voltage cannot improve the resolution of the OSSPSU motor.

According to this comment, we added the following statement in the revised manuscript:

“To ensure that the piezoelectric materials are not depolarized, in this work, the maximum AC driving voltage applied on the crystal-OSSPSU and ceramic-OSSPSU motors are set to be 150 V_{P-P} and 250 V_{P-P} , respectively.”

Fig. R8 The output performance of the OSSPSU motors. **a**, The relationship between the bidirectional motion velocities of [001]-PIMNT crystal-OSSPSU motor and the applied voltage and load; **b**, The relationship between the bidirectional motion velocities of hard ceramic-OSSPSU motor and the applied voltage and load; **c**, The step displacements of [001]-PIMNT crystal-OSSPSU motor as a function of the applied voltage; **d**, The step displacements of hard ceramic-OSSPSU as a function of the applied voltage.

Response to reviewer #3

Comment: The authors present a comprehensive work on design strategy of metamaterial with non-zero piezoelectric coefficients with supporting FEM simulations.

Reply: We kindly thank the reviewer for considering our manuscript as “a comprehensive work on design strategy of metamaterial with non-zero piezoelectric coefficients with supporting FEM simulations”. While, we would like to note that the design of non-zero piezoelectric coefficients by orderly stacked structure with piezoelectric strain units (OSSPSU) via FEM simulations is only a portion of our present work. With our proposed OSSPSU, in addition, we achieved ultra-high strains and artificially coupled multi-vibration modes. To verify the feasibility of the OSSPSU, we designed a coupled mode OSSPSU stator and successfully fabricated a standing wave piezoelectric motor, which is one of the smallest piezoelectric motors ever reported. The motors showed great advantages in velocity and step resolution when compared with state-of-the-art motors. Therefore, our present work is promising to greatly benefit not only the design of metamaterials with all non-zero piezoelectric coefficients but also the development of advanced electromechanical devices.

Comment: The reviewer is however unable to appreciate the novelty of this work given similar earlier publications like <https://www.science.org/doi/full/10.1126/sciadv.aax1782>. Hence, I find it not suitable for publication in Nature communications.

Reply: Thanks for the comment and also thanks for sharing the reference, i.e., Ref. 31 in our manuscript. We are aware of the contribution of Dong *et al.* (Ref. 31) on the design of piezoelectric metamaterials for optimizing piezoelectricity. However, we respectfully disagree with the statement that “the novelty of this work is similar to earlier publications”. The design principle of piezoelectric metamaterials and electromechanical devices designed in our present work are very different from those reported in previous works. The novelty and advances of our work are listed as follows.

- (1) We designed the metamaterials with all non-zero piezoelectric coefficients based on the uniform strain unit and the same boundary condition of a topological structure, while those are not totally unified in previous works. For example, in Ref. 31, the non-zero piezoelectric coefficients d_{14} and d_{11} were achieved by triangle and square units, respectively. This feature of our present work, i.e., unified strain units and boundary condition, can contribute to a broader range of electromechanical devices from piezoelectric single vibration mode to coupled multi-vibration modes.
- (2) In our present work, the OSSPSU was successfully utilized to design ultrasonic motors, and the newly fabricated motor is one of the smallest ultrasonic motors ever reported. Owing to the ultrahigh strain of the artificially coupled vibration modes, the motor possesses high speed in unit volume of $4.66 \text{ s}^{-1} \text{ mm}^{-2}$, and its moving resolution is $\sim 3 \text{ nm}$, being two orders of magnitude higher than that of state-of-the-art piezoelectric motors.
- (3) We would like to emphasize again that the design of metamaterials with nonzero piezoelectric coefficients is just one objective of our present work. The main achievements of our work also include the realization of artificially coupled multi-modes in piezoelectric metamaterials and the

design of ultra-high performance ultrasonic motors, which are not involved in previous literature.

According to this comment, we made additional discussions in Introduction part of the revised manuscript, in order to show the difference and significance of our present in contrast to earlier publications, as listed below:

“Recently, it has been reported that all nonzero and ultrahigh piezoelectric coefficients can be realized in piezoelectric materials based on the construction idea of metamaterials^{30,31}, inspiring the design of electromechanical devices. However, many piezoelectric devices, such as ultrasonic motors, require coupled vibrational modes for high electromechanical conversion capability. In previous reports, the coupled vibration modes and the construction of all nonzero piezoelectric coefficients aren’t achieved simultaneously, since the different piezoelectric strain units and/or different boundary conditions are adopted for inducing different piezoelectric vibration modes. For example, as reported in Ref. 31, the non-zero piezoelectric coefficients d_{14} and d_{11} were achieved by triangle and square strain units, respectively. Thus, to fulfill the requirements of piezoelectric devices, the metamaterials that can achieve all non-zero piezoelectric coefficients and coupled vibration modes by the uniform piezoelectric strain unit and boundary condition are highly desired.”

REVIEWERS' COMMENTS

Reviewer #2 (Remarks to the Author):

This paper has been revised according to the comments of reviewers and it is good enough for formal publication nowadays.

Response Letter to Reviewers' Comments

Response to reviewer #2

Comment: This paper has been revised according to the comments of reviewers and it is good enough for formal publication nowadays.

Reply: We thank the reviewer for his/her positive recommendation and his/her efforts on reviewing the manuscript.